# How the War in Ukraine Affects Food Security

**DOI:** 10.3390/foods12213996

**Published:** 2023-10-31

**Authors:** Walter Leal Filho, Mariia Fedoruk, João Henrique Paulino Pires Eustachio, Jelena Barbir, Tetiana Lisovska, Alexandros Lingos, Caterina Baars

**Affiliations:** 1European School of Sustainability Science and Research (ESSSR), Hamburg University of Applied Sciences, 21033 Hamburg, Germany; walter.leal2@haw-hamburg.de; 2Department of Natural Sciences, Manchester Metropolitan University, Manchester M1 5GD, UK; 3Research and Transfer Centre Sustainability & Climate Change Management (FTZ-NK), Faculty of Life Sciences, Hamburg University of Applied Sciences, 21033 Hamburg, Germanytetiana.lisovska@haw-hamburg.de (T.L.);

**Keywords:** food security, Ukraine, war, agriculture, economics, public policies

## Abstract

The war in Ukraine has caused severe disruption to national and worldwide food supplies. Ukraine is a major exporter of wheat, maize, and oilseeds, staples that are now suffering a war-triggered supply risk. This paper describes the background of the problem and illustrates current trends by outlining some of the measures that may be deployed to mitigate the conflict’s impacts on achieving SDG 2 (Zero hunger), especially focusing on ending hunger, achieving food security, improving nutrition, and promoting sustainable agriculture. In order to understand the main research strands in the literature that are related to food security in the context of wars, the authors adopted a bibliometric literature review based on the co-occurrence of terms technique, conducted with 631 peer-reviewed documents extracted from the Scopus database. To complement the bibliometric assessment, ten case studies were selected to narrow down the food insecurity aspects caused by the war in Ukraine. The co-occurrence analysis indicated four different thematic clusters. In the next stage, an assessment of the current situation on how war affects food security was carried out for each one of the clusters, and the reasons and possible solutions to food security were identified. Policy recommendations and theoretical implications for food security in the conflict context in Ukraine were also addressed.

## 1. Introduction

The armed conflict in Ukraine poses a considerable challenge to food markets worldwide. On 17 July 2023, the UN Secretary-General, Antonio Guterres, noted the following: “The production and availability of food is being disrupted by conflict, climate change, energy prices and more”. These words show that addressing food security is complicated by conflicts, including active military operations in the territory of Ukraine, climate change, and energy prices [1]. In accordance with the most recent definition by FAO [2], Food Security (FS) has been defined as follows: “*A situation that exists when all people, at all times, have physical, social and economic access to sufficient, safe and nutritious food that meets their dietary needs and food preferences for an active and healthy life”.*

Before 2014, humanity was successfully overcoming the problem of food security in small steps; there has been a continuous decline in food security indicators since then. The impact of the COVID-19 pandemic on food security has also been devastating, with an estimated increase in the number of people suffering from hunger in 2020 equal to the previous five years combined. This means that almost one in three people in the world (2.37 billion) did not have access to adequate food in 2020 [3]. The FAO’s 2023 report notes that the international community has not had time to recover from the global pandemic and is now struggling with the consequences of the war in Ukraine, which has further undermined food and energy markets [4]. Global food security challenges and its drivers include conflicts and wars in Ukraine and other countries, slowdowns and downturns, and climate change [4]. 

Since February 2022, the war in Ukraine has caused numerous consequences, including severe stress in food security, with impacts on the national and global levels. The reasons for this severe stress are rising prices and supply chain disruptions [5]. Most importantly, Ukraine is one of the world’s main breadbaskets [6]. Ukraine exports approximately 20% of all wheat produced in the world. As for corn and sunflower, Ukraine accounts for 10% and 45% of global production, respectively [7].

Ukraine is among the largest exporting countries of sunflower seeds, oilseeds, rapeseed, corn, and wheat [8]. For example, Ukraine accounts for about a third of cereal supplies in the Middle East and North Africa region [9]. East Africa is also heavily dependent on cereal imports, 45 per cent of which come from Ukraine [9]. Fragile countries such as Libya, Pakistan, Yemen, and Lebanon receive 30% of their wheat imports from Ukraine [2], indicating that a large percentage of the population lives in food insecurity.

In 2021, Ukrainian farmers grew the largest harvest in agri-production history; as a result, the exports to the EU increased by 50%, leaving behind the trade with China and Russia [10].

According to the results from Table 1, the harvesting of cereals and oil crops in Ukraine increased overall during the time frame 2018–2021, for all presented categories except soya beans. In order to better understand the relative differences in production volumes, the dynamics of cereal and oil crop production are shown in Figure 1.

There was a decrease in harvesting for all presented agricultural holdings in 2020 (Figure 1); however, 2021 was very productive overall and especially for maize.

The largest importers of wheat from Ukraine are Egypt, Indonesia, Turkey, Pakistan, Bangladesh, Tunisia, Morocco, Yemen, Lebanon, Philippines [12].

In the first 8 months of 2021, Ukraine exported products of the oil and fat industry that were worth USD 4.1 billion, which is 11% more than in the same period in 2020. This information was provided by the Deputy Minister of Economy of Ukraine, Taras Kachka, in an official announcement on the ministry’s website [13].

Before the war, Ukraine provided 46% of the sunflower oil exports, 9% of wheat exports, 17% of barley exports, and 12% of corn exports to the world markets, according to data from the US Department of Agriculture [14]. Despite the hostilities, food manufacturers are resuming work in the de-occupied territories, introducing new capacities in the western and central regions. The industry is restrained by export restrictions, a reduction in the purchasing power of the population, and an increase in the cost of production due to logistical difficulties and an increase in the price of energy carriers. However, some manufacturers maintain production volumes, reorienting themselves from the Ukrainian to the European market [15].

As of 30 June 2022, the Agricultural Price Index was 34% higher in comparison to January 2021. Maize and wheat prices were 47% and 42% higher, respectively, when compared to January 2021, while rice prices are about 8% lower. According to the World Bank’s April 2022 Commodity Markets Outlook, the war in Ukraine has altered global patterns of trade, production, and consumption of commodities in ways that will keep prices at historically high levels through the end of 2024, exacerbating food insecurity and inflation [16].

Food prices were already high before, and the war is driving food prices even higher (Figure 2). Commodities that have been most affected are wheat, maize, edible oils, and fertilizers. Global commodity markets face upside risks through the following channels: reduction in grain supplies, higher energy prices, higher fertilizer prices, and trade disruption due to the shutting down of major ports [16].

International wheat prices had a steep increase in May 2022 in response to the announcement of a wheat export ban by India. As exports from Ukraine remain hindered by war disruptions, and after India’s larger wheat shipments in recent months played an important role in partially offsetting lost exports from Ukraine, the country’s export restriction exacerbated global availability uncertainty. According to the UN FAO, the disruptions caused by war in Ukraine have raised the average food price index to its highest level ever (Figure 3). Thus, the FAO Food Price Index, which tracks international prices for the world’s top-selling food commodities, increased by an average of 14.3% in 2022 compared to 2021, the highest since records began in 1990. The FAO Cereal Price Index in 2022 increased by 17.9% due to factors such as significant market disruptions, rising energy and input prices, adverse weather conditions, and continued high global food insecurity, FAO summed up [17].

The war has significantly disrupted the transport of commodities. Almost all of Ukraine’s grain exports went through Black Sea ports, which were not operational before the “Initiative on the Safe Transportation of Grain and Foodstuffs from Ukrainian ports” was signed in Istanbul, Turkey, on the 22nd of July 2022. It was expected that Ukraine would export up to 20 million tonnes of wheat during the season (summer 2022), which corresponds to about 10% of global wheat exports.

However, in 2023, the situation remains critical. According to information from the Ministry of Foreign Affairs of Ukraine, in the first 10 days after the breakdown of the Grain Deal, shelling launched towards Ukrainian territory destroyed 180,000 tonnes of grain and damaged 26 port infrastructure facilities and five civilian vessels [18].

The war will also disrupt agricultural production in Ukraine in the coming seasons. Spring planting for crops such as maize, barley, and sunflowers typically occurs from April to May, while winter wheat is planted from September to mid-November. Shortages of labor and inputs (such as fuel and fertilizers), destruction of farming equipment, and safety concerns of growers will have a severe impact on Ukraine’s agricultural (and especially wheat) production. Estimates on how much Ukraine’s agricultural production will decline in the upcoming season vary from 25 to 50 percent [19].

In May 2022, it was estimated that one-third of Ukrainians suffered from food insecurity [12]. Additionally, with the beginning of the crisis, the number of ships carrying agricultural products dramatically declined, which in turn resulted in considerable impact on food prices [12]. Furthermore, the ripple effects of the war encompassed Ukraine’s closure of harbors and interruptions of the oilseed crushing process [2]. That is why a shortage of food exports in Ukraine is accompanied by soaring food prices [12].

Due to the war in Ukraine, ripple effects have led to record levels of food insecurity. The number of people experiencing acute food insecurity or at high risk as of June 2022 (million) increased by 7.3 in Asia and the Pacific, 6.3 in the Middle East, North Africa, and Eastern Europe, 9.7 in West Africa, 9.6 in Southern Africa, 10.5 in Eastern Africa, and 3.6 in Latin America and the Caribbean [20].

However, in order to meet the current demand for wheat, 25 million tonnes need to be replaced by alternative sources [21]. At COP 27, The “Green Grain Paths” Ukrainian initiative was presented, which raised the issue of the global threat to the world’s food security and the creation of partnerships at every stage of the grain supply chain.

The FAO’s 2023 report predicts that nearly 600 million people will be chronically undernourished in 2030, indicating a huge gap before the SDG goal of eradicating hunger is met. By comparison, this figure is about 119 million more than in a scenario where there was neither a pandemic nor war in Ukraine, and about 23 million more than if there had been no war in Ukraine [22].

The ongoing fighting in Ukraine could trigger a global food crisis, as 36 of the 55 countries already experiencing a food crisis depend on exports from Ukraine and Russia. Before 2022, both countries produced about 30% of the world’s supply of wheat and barley, a fifth of corn, and more than half of sunflower oil [23]. Roughly 22% of Ukraine’s farmland—including 28% of winter crops and 18% of summer crops—is not under Ukrainian official control, according to NASA Harvest’s analysis [24]. The “Initiative on the Safe Transportation of Grain and Foodstuffs from Ukrainian ports” (Black Sea Grain Initiative) stabilized spiraling food prices around the world and prevented hunger affecting millions of people [25].

Due to this reliance on exports from both countries, the war in Ukraine is affecting food security globally. The concept of food security encompasses four dimensions: food availability, access, stability, and utilization. The addition of a further dimension to the concept, like sustainability or agency, has been discussed among scholars [26] and has been recognized as an official dimension by the FAO since 2022. Food insecurity has various consequences—whether for individuals and those suffering from hunger or malnutrition, which leads to physical and mental health problems that might affect the economy and cause poverty, or for food security itself, causing more conflicts due to the strain it puts upon people [27]. Therefore, closing knowledge gaps on mitigating food insecurity is urgently needed in light of the Ukraine situation. Furthermore, it has high importance in helping the world reach higher strategic goals like SDG2 or SDG16.

To conclude, the war in Ukraine has led to severe disruptions in terms of agricultural production [12]. The Black Sea Grain Initiative (BSGI) along with the UN Memorandum of Understanding with the Russian Federation to facilitate Russian food and fertilizer exports have become lifelines for global food security and the stability of world food prices. These measures helped to bring food prices down by 23 percent from their maximum in 2022 [28].

Based on previous experiences and studies, the war itself can have long-term influences on the physical, chemical, and biological characteristics of soil, and using this land for growing food might potentially lead to health problems [29]. In this sense, apart from describing the background of the problem, this study has two main goals: (1) to unveil what the literature has been addressing in food insecurity in the context of war, and (2) to cross-check the theory against some current trends and data, assessing the impact of the ongoing conflict on agricultural production in the affected area, and consequently, estimating the impact on food availability.

In order to achieve these goals, the authors opted for a bibliometric analysis of the literature, where a search string was created to understand what scholars have been covering in the field. After a screening process, a total of 635 peer-reviewed papers were selected and used to proceed with the co-occurrence technique. In the second stage, the authors judiciously selected nine case studies to illustrate how the impact of war in Ukraine puts food security in danger on local and global scales. This also allowed us to indicate measures that may be deployed to mitigate the conflict’s impacts on achieving SDG 2 (Zero hunger), with a focus on ending hunger, achieving food security, improving nutrition, and promoting sustainable agriculture.

## 2. Methods

The present research considers a bibliometric analysis and the case study research approach developed from a literature review [30,31], as it is relevant to gathering broad evidence on the impacts of wars on food security. The approaches can then also be used to analyze the narrower context of the conflict in Ukraine.

In this context, the bibliometric analysis was conducted for specific terms related to war and food security on Scopus, a widely known scientific database of peer-reviewed academic documents containing more than 5000 journals and 24,000 publications. The bibliometric analysis is considered a quantitative computer-assisted review methodology commonly used to identify the main studies, authors, and journals or to understand the bibliometric networks evidencing, for example, the major discussions of a research field [32,33].

The data collection took place in August 2022, where the documents used to compose the sample were collected through a search string containing terms connected to ‘war’ and food security, such as ‘food’ OR ‘grain’ OR ‘cereal’ OR ‘agriculture’ OR ‘crop*’ OR ‘agricultural’. Figure 4 shows how the authors worked with the exclusion criteria before moving forward with the bibliometric analysis.

The search string initially returned 732 documents and, after applying the exclusion criteria, as stated in Figure 4, 631 documents remained and were used to proceed with the co-occurrence analysis. Table 2 shows the search string used and the number of entries obtained to proceed with the bibliometric analysis.

The investigation was performed using the VOSviewer software [34], and the co-occurrence of terms was the technique selected to explore the research streams that researchers have been discussing on food security in the context of war. The results are presented in the next section in a network graph (see Figure 5), where the diameter of the bubbles indicates the frequency of the occurrence of a specific term, while the link width indicates the strength of the connection between two terms. The terms that appear close to each other are related to the frequency of co-occurring. Therefore, they are expected to be associated, generating a thematic cluster due to their co-occurrence frequency.

In this sense, the bibliometric analysis becomes relevant for this study for several reasons. For example, the bibliometric analyses based on the co-occurrence of terms could help scientists to identify research trends in a specific field, evidencing its big picture, as well as helping the researcher to map the knowledge of each one of the clusters found in order to delve into a specific area of the studied field. Therefore, the co-occurrence technique could help scientists to reduce the complexity of a field, supporting the conduct of discussions based on how different concepts converge and relate to each other (see [32,33,35].

This paper also considers a set of 10 case studies to narrow the discussion on the impact of war on food security in the Ukrainian context. The case study selection was based on the expert-driven literature review to find relevant well-cited papers that are published in top journals and that could enhance the discussion on the field as well as tackle the goals of this study. The selection of case studies was judiciously conducted in order to overcome some limitations of the bibliometric analysis and deepen the discussion of food insecurity aspects caused by the war in Ukraine as well as guaranteeing thematic relevance, variety of scopes, and international interest. It is worth considering that the case studies are decoupled from the bibliometric analysis, being a second methodology used in the study. In this sense, this approach not only enhanced the study’s robustness and credibility, but also tackled potential constraints of quantitative methods such as the bibliometric analysis.

Finally, this study also relied on secondary data to understand the impact on food availability in the affected area. In this sense, the authors gathered relevant information from relevant organizations such as the Food and Agriculture Organization of the United Nations [2], the European Commission [9] World Food Programme [12], The World Bank [16], and some Ukrainian statistics divisions to measure and understand the magnitude of the impact of the shortage of grains due to the war.

## 3. Results and Discussion

### 3.1. Food Security in Times of War

As an outcome of the bibliometric analysis, four different clusters (C) were identified and used in order to assess the current situation on how war affects food security (Figure 5).

C1: The red cluster focuses on the social dimensions of the interlinkage between food security and war. War, as one of the main causes of food insecurity, causes destruction, displacement, control of food, and starvation, which makes governments often unable or reluctant to address food shortages, and instead they try to prevent the exacerbation of the situation [36]. During and after war, many veterans and displaced people, especially those with a higher duration of displacement, experience food insecurity and low nutritional status [37,38]. In a study from Widome et al. [37], veterans with a lower income and military pay grades were more likely to smoke and drink alcohol, and they slept less than food-secure humans.

Moreover, trade wars or international food trade barriers due to political conflicts lead to food insecurity [39]. As a result, countries banning food trade experience a surplus of food, leading to an increase in the global environmental cost of agricultural production [40]. Other results include food distribution inequality, violation of basic human rights, and the maintenance of social disorder [41].

Internal factors such as economic recession, environmental deterioration, political instability, and inter-ethnic conflicts are other reasons for food insecurity [39]. Famine manifestations in people can include nutritional deficiency, low weight, anemia, delay in children’s physical development, and childhood diseases. Vitamin deficiency is also one of the main deficiency-related diseases among food-insecure people [42]. Reduction in the production of fruits and vegetables results in a dramatic rise in prices, a reduction in human consumption, and an increase in human vitamin C deficiency [43]. As far as it leads to nutritional deficiency, there is a clear connection to C2 (human history, nutrition, and war). It is important to note that every type of war can affect food security in a different manner, depending on the extent of the region affected and of the destruction [44].

Suggestions made by authors in this cluster to address food insecurity at a global level include respecting food as a basic human right, merging local capacities [35], creating a sustainable global cropping system [39], and more equitable seed distribution to rehabilitate agriculture disorder [41]. The government’s measures to address food insecurity at the national level include overseas agricultural investment, food import from multiple origins [45], food independence by replacing available foods with scant foods [40], and taking initiatives aimed at offering people the healthiest diet possible during food shortages [42]. The nationalization of agricultural land is considered an impractical way to address urgent food shortage problems [46].

C2: The green cluster emphasizes the nutritional dimension of food security and war, showcasing historical examples.

The analysis presented in the paper is closely connected to the historical context and empirical evidence provided in these excerpts, which underscores the critical importance of addressing food security in conflict contexts [47]. These historical references to World War I and World War II serve as poignant reminders of the devastating consequences of food shortages during conflicts, setting the stage for understanding the contemporary challenges faced by Ukraine in the context of its ongoing conflict [48,49].

Moreover, the information regarding the direct and indirect impacts of war on agricultural crop production offers valuable insights into the complexities of the issue [50]. The destruction of infrastructure is highlighted as a direct cost of war, resulting in a tangible reduction in crop production. Additionally, the contamination of food and the environment with substances such as uranium, which is used in bullets and shells during wars, underscores the far-reaching consequences of warfare on food safety and quality [51]. This multifaceted analysis contributes to a comprehensive understanding of the challenges Ukraine faces in ensuring food security amidst conflict.

Furthermore, the focus on the nutritional deficiency and health consequences of food insecurity during and after wars aligns perfectly with the broader discussion on achieving Sustainable Development Goal 2 (Zero Hunger) [52] and interconnects with C3 (agriculture, food safety, and health). The references to studies detailing the adverse effects of wartime food shortages on women’s health, child development, and veterans’ well-being emphasize that addressing food security during conflict extends beyond immediate hunger relief; it has profound and lasting health and social implications [37,42,53,54].

Lastly, these insightful excerpts are highly relevant to the paper’s policy recommendations. By highlighting both historical precedents and contemporary evidence of the wide-ranging impacts of food insecurity during conflicts, the paper can make a compelling case for specific policy actions. For instance, it may recommend strategies to protect agricultural infrastructure, mitigate the environmental impact of war, and address nutritional deficiencies among affected populations [46]. In essence, these excerpts, rich with references, provide crucial background information and empirical evidence that support the paper’s analysis of food security in the context of the conflict in Ukraine, enriching the overall discussion and policy recommendations.

C3: The blue cluster represents the complex relationship between agriculture and food production with food availability, and, consequently, their impacts on human health and well-being. Other factors addressed in this cluster are the industrialized world and the impacts of climate change, along with wars, displacement of people, and agricultural land abandonment.

Generally, developing countries depend on agriculture as a primary driver of the economy as well as the main provider of nutrition for people. However, when a military conflict takes place in a country, it usually leads to disruption in agricultural production, primarily in lands closest to the war areas [55]. Farmers fear for their lives and might abandon their land until the war is over (Scopus, Vulnerabilities of Wheat Crop Farmers in War Zone, n.d.), while also facing psychological stressors and financial corruption [56]. The conflict may cause a change in the agricultural behavior of the population, which may disrupt the ecological system and threaten wildlife without creating an actual increase in crop production or food security [57].

There are also the issue that various kinds of weapons used in war zones leave behind the harmful waste of heavy metals or even nuclear waste. This leads to damage such as irreversible pollution. As far as it results in the contamination of food, there is a clear connection to C2 (human history, nutrition, and war). Moreover, it leads to the destruction of the ecosystem and animal life such as insects, which are important for pollination and crop production in those areas [58,59]. Along with the effects of war, global climate change may also contribute further to food insecurity [60].

The cluster also shows efforts and solutions to agricultural and ecological crises during and after war and emphasizes the importance of thorough research regarding all aspects related to the targeted area, including the geographical, ecological, and ideological factors to provide the ultimate, long-lasting solutions [61]. This could include enhancing agriculture in safe zones [55], encouraging farmers to use crops that fit the nature of their land and guarantee maximum production and food safety as much as possible [60], or training staff on multiple levels to upgrade agricultural production [61]. Moreover, during a war situation, satellites could be used to alleviate food insecurity in the target country, facilitating food aid by distributing it where it is most necessary [55].

C4: The yellow cluster illustrates the impact of war and political conflict on various aspects of the economy and vice versa and how this shapes the public policies in politics, health, education, and other domains.

For instance, it was observed that poverty and low wages in developing countries push farmers and workers in the agricultural field to join the armed conflict due to their constant economic struggle [62]. In addition, the failure of policies and the market system in certain economies played a major role in igniting the war in these countries, leading to destruction and further economic suffering [63]. On the other hand, armed conflict always has a devastating impact on the economy; all aspects of national production are affected, including agriculture [50], and as a result, many people within the population lack the essential food components for sufficient nutrition, in part due to the high prices that accompany food shortages [64,65].

Another phenomenon is that the economy and international trade, especially food trade, have been a growing cause of political dispute [66]. A good example of it could be market spoilage. At the same time, some economic decisions have been made as a result of political disputes, which consequently puts the food and nutrition security in some countries in jeopardy [40]. However, there have been some great efforts from various international organizations to guide public policies and international public relations towards a more fair trade of food in order to help countries that have suffered from war to have a certain level of nutrition security as a fundamental part of human rights [67,68]. On the same level, local and national economic policies in post-war countries play a major role in shaping the future and maintaining peace by implementing successful practices of sustainability and productivity [63].

### 3.2. Food Security in Conflict Context in Ukraine

When a military conflict takes place in the country, multiple factors have to be considered. Therefore, the impact of war on food security in the Ukrainian context would be different compared to other conflict situations in the world. In order to understand the situation of food insecurity aspects caused by the war in Ukraine, an analysis of 10 case studies was conducted. These case studies were selected based on the expert-driven literature review. The results of this study are presented below.

CASE STUDY 1: “Impacts of the conflict in Ukraine on global food security” [69].

Short description

In particular, the study examines the possible impact of the war in Ukraine on Ukraine’s production and export of agricultural commodities, especially grains and oilseeds. It also outlines the impact of trade sanctions against Russia on the export of cereals and oilseeds. At the same time, this research investigates the possible impact of higher energy prices on global agricultural production and trade.

Implications for Food Security

The study concludes that from a food security perspective, there is enough food on the global level. However, higher food prices could become a problem and put food availability under some pressure for a part of the population that has a low income, spends a large part of their income on cereals, and is highly dependent on imports of Ukrainian and Russian cereals (for example in Egypt, Turkey, and the Middle East). For the EU, food prices rose by 5% for crops and 2% for food; however, given the relatively low proportion of income spent on food, this should not be a problem for the average citizen.

CASE STUDY 2: “Implications of the Russia–Ukraine war for global food security” [70]

Short description

This research investigates how conflicts can affect the capacity of food systems and supply chains to function appropriately due to production declines (producers are being engaged in war, are unable to produce, or are fleeing the country), destroyed agricultural yields and water infrastructure, or disruptions in agricultural inputs in the foreign markets, as well as how these can affect the capacity of consumers to access sufficient food because of their declining purchasing power. At the same time, the paper outlines what happens with the capacity of international food aid when trying to meet growing food needs in times of crisis.

This study suggests solutions to listed problems, such as establishing an international community strategic food reserve or including new rules in international humanitarian law that provide sufficient protection to food systems-related infrastructures and activities.

Implications for Food Security

After a decline during the past decade, global hunger is rising again, and the ongoing war in Ukraine is expected to increase this trend (experts estimate that 7.6 to 13.1 million people are threatened).

The reason for this is the volatility of major food commodities and fertilizers, which may affect production decisions and spur speculative behavior. It will affect international society, but countries most dependent on imports of Ukrainian and Russian cereals, such as Egypt, Lebanon, and Tunisia (whose wheat imports from Ukraine are 85%, 81%, and 50% of their total wheat imports, respectively), will be the most impacted by war-related disruptions.

This war is also affecting the ability of international agencies to provide food aid to countries that are suffering from famine or other armed conflicts because of rising costs, with the risk of excluding millions of people from current food aid programs.

CASE STUDY 3: “Potential medium-term impacts of the Russia-Ukrainian war on the Dutch agriculture and food system: An assessment” [71]

Short description

This research assesses the potential impacts of the war in Ukraine on different agricultural sectors in the Netherlands, both with and without taking into account the European Union or national policy response. The focus is on the medium-term impacts (period 2021–2025), considering 2021 as a pre-Ukraine war reference point. In this study, the authors also explore potential impacts on employment. The assessment addresses the impacts on primary production, as well as on upstream and downstream industries. The results show the impact on the Netherlands’ use (including consumption) and on the Dutch agricultural net trade position.

Implications for Food Security

The impacts found as a result of this research for the different scenarios suggest that the medium-term impact of the Ukraine war on Dutch agriculture is relatively limited (less than 1% of the total value added over the period 2022–2024 in the five considered sectors). Moreover, energy prices and agricultural commodity prices were already increasing because of other factors (the COVID-19 pandemic, the role of droughts, etc.).

At the same time, the uncertainty with respect to energy prices and energy-related inputs (fertilizer, pesticides) is likely to be one of the most pronounced impacts of the Ukrainian war on Dutch agriculture in the short to medium term. An energy price increase is likely to have worldwide impacts on agricultural production and consumption and will create a passing on effect on agricultural product prices.

CASE STUDY 4: “Quantifying War-Induced Crop Losses in Ukraine in Near Real Time to Strengthen Local and Global Food Security” [72]

Short description

In order to estimate the direct and indirect effects of the war and the expected yield of winter crops in Ukraine, a 4-year panel (2019–2022) of 10,125 village councils in Ukraine was used. Satellite imagery (by Sentinel-2, Planet) was used to provide information on direct damage to agricultural fields (burning, the explosion of ordnance, rockets, or aircraft, and soil compaction) to classify crop cover using machine learning, and to compute the Normalized Difference Vegetation Index (NDVI) (by sensors Landsat and MODIS) for winter cereal fields as a proxy for yield. This research focuses on the supply side in order to investigate the effects of the war in Ukraine on food security, unlike the other studies that focus rather on the demand.

Implications for Food Security

The result of this study shows that food production can be analyzed using satellite imagery to generate measures of agricultural production as outcome variables as well as indicators of conflict, considering that causality runs from conflict to food security (opposite to what is most prevalent in developing countries, that the value of agricultural endowments triggers conflict). Particularly in Ukraine, taking area and yield reduction together suggests a war-induced loss of winter crop output of up to 25% if the current winter crop can be harvested fully. Having this information can help to improve decision making by policymakers and private parties to minimize war-induced losses.

CASE STUDY 5: “The conceptual principles of state policy of Ukraine in the field of food security in terms of European integration” [73]

Short description

The article (1) examines organizational and economic features (problems) of insufficient food supply in Ukraine; (2) substantiates the structural components of the program of balanced production of food and food raw materials; (3) determines measures of state policy on the food security of the population of Ukraine; (4) develops a methodological approach to substantiation and implementation of state policy on the implementation of the principles of ensuring food safety and quality in Ukraine; (5) substantiates the strategy of ensuring the physical and economic accessibility of food.

Implications for Food Security

As a result of this research, the structural components of the program of balanced production of food and raw materials were formed (increasing the level of income, providing cheap food, increasing the supply of food, and the use of special tools). The mechanism of the methodological approach construction is formed on the principle of problems–directions–measures. The proposed strategic tools (structural, special, and local measures) enable solutions for the tasks of improving nutrition at the local level and gaining additional positive impact on the formation of a model of sustainable agricultural development, which can influence global food security.

CASE STUDY 6: “The reinvasion of Ukraine threatens global food supplies” [74]

Short description

This study outlines that the reduction in the vast quantities of edible food that are wasted in rich countries every day, as well as the replacement of animal-based foods with plant-based options, could feed an additional number of people and mitigate climate change. It also questions the existing model of global trade and systems of agricultural production, which is dependent on oil for fertilizers, transport, and agrichemicals; this model encourages the consumption of energy-dense ultra-processed foods.

Implications for Food Security

According to this research, the war in Ukraine, together with pre-war factors (e.g., COVID-19), caused the already rising global food prices to reach an all-time high. The consequences will be felt by everyone, but some populations are especially vulnerable (Lebanon, Yemen, Syria, Afghanistan). There are also many poor people in rich countries who will suffer. Several actions are required in order to combat food insecurity, such as replacing all animal-based foods with plant-based ones, which could feed an additional 350 million people, and changing the existing model of global trade in order to increase food stocks in countries that have disinvested in domestic production in favor of cheaper imports. At the same time, the systems of agricultural production need to be less energy-dense.

CASE STUDY 7: “The War in Ukraine, Agricultural Trade and Risks to Global Food Security” [75].

Short description

This research investigates global food supply chains. The war in Ukraine, together with rising prices in late 2021, have affected import-dependent countries in the Middle East and North Africa (MENA) region and sub-Saharan Africa, which rely heavily on Russian and Ukrainian wheat. However, global demand for wheat is expected to be met in the current marketing year (2022/23), since countries such as Australia, Brazil, and the USA will increase exports to fill the gap left by Russia and Ukraine.

Implications for Food Security

According to this research, the war in Ukraine increases the risk of food insecurity for import-dependent countries with low per capita incomes. The key to dealing with crises and mitigating the risks of food shortages could be a reduction in bureaucratic and tariff barriers to trade by global supply chain structures. Also, targeted political efforts are needed to ensure that Ukraine and Russia remain integral parts of the world agricultural trading system. On the other hand, transitions to (more) closed food economies are not advisable, as this would remove players from international markets, could lead to food shortages in many countries, and might take focus away from environmental and health-related issues.

CASE STUDY 8: “We need a food system transformation—In the face of the Russia-Ukraine war, now more than ever” [76].

Short description

This paper discusses moving towards changes in global food consumption habits and the food supply chain. The research shows how transformation towards a healthy, fair, and environmentally friendly food system reinforcement can bring short-term relief and avert the existential threat our food system poses to the health of people and the planet. It confirms that the need for sustainable solutions is still present while facing the Russia–Ukraine war.

Implications for Food Security

The authors of this study suggest that global food insecurity has its origin not in a shortage of supply but in high economic inequalities and maldistribution. Even though global food production is more than sufficient to feed an even higher world population, it is not supplied to those with limited financial means. This paper proposes three levers for solving short-term problems and long-term sustainable development: 1. Accelerate the shift toward healthier diets with fewer animal products in Europe. 2. Increase the production of legumes and strengthen Farm2Fork. 3. Reduce the amount of food waste. “Farm to Fork” is, namely a scheme led by the European Union, to promote the sustainability of farming processes, i.e., from production (the farm) until it reaches the plate of the end-consumer (the fork).

CASE STUDY 9: “What the war in Ukraine means for energy, climate and food” [77].

Short description

This paper shows how war in Ukraine influenced the markets and geopolitics of energy, oil, and gas prices and the reconsideration of countries’ energy supplies. Particularly, it outlines the possibility of rising energy prices and the potential loss of grain supplies from Ukraine and Russia to reinforce inflationary effects and drive up prices for food and other commodities.

Implications for Food Security

According to Christopher Barrett, in the short term, the prices of wheat and other grains have increased due to hoarding and bidding wars. But global food stocks are sufficient to cover the loss. Even though there could be disruptions to fertilizer markets because fossil fuels are a major feedstock, the effect should be mitigated by substitutes. However, the cost of rising petrol and electricity prices to the larger food supply system could be substantial. As a result, some of the greatest casualties of the war in Ukraine will be people who are already severely struggling in other places (Yemenis, Syrians, and Nigerians).

CASE STUDY 10: “Caught off guard and beaten: The Ukraine war and food security in the Middle East” [44].

Short description

The study provides a detailed analysis of the food-related vulnerability of the Middle East (Yemen, Libya, Lebanon, Sudan) in the wake of the Ukraine war. It shows the varying impacts of it, including a deepened food sector crisis, worsened by political–economic instabilities, limited domestic agriculture, and the lack of reliable grain reserves. At the same time, the paper highlights country-level response strategies related to regional aid and cooperation that have emerged in the Gulf countries.

Implications for Food Security

The author of this study outlined that the war in Ukraine is hitting the Middle East very hard due to the relatively high dependence levels and the baseline political, economic, and environmental vulnerabilities. For example, around 80% of the 30 million Yemeni population is dependent on aid. At the same time, countries having significant storage capacity and available reserves were better able to avoid or delay shortfalls or price hikes despite high dependence (e.g., the Gulf countries and Egypt). As a short-term solution, food subsidy systems for the most vulnerable have softened some of the impacts of the price hikes. In oil- and/or gas-exporting countries, the increase in carbon fuel prices after the war in Ukraine meant that additional revenues could be used in the mitigation of the food crisis (e.g., in Algeria, Libya). At the same time, environmental vulnerabilities can weaken the ability of local agriculture to provide food. As a result, many Middle East countries, especially with political–economic instability (Yemen, Libya, Lebanon, Sudan), will be relying on international cooperation in the short and medium term, including shipments under the Black Sea Grain Initiative.

It should be noted that the case studies illustrate a variety of situations and contexts where food security is endangered because of the ongoing war in Ukraine or is avoided due to various means used to achieve it. These are summarized in Table 3.

According to the results obtained by the analyses of a set of 10 case studies, the war in Ukraine can affect food security due to many reasons. Among them are production declines (producers are being engaged in war, unable to produce or flee the country), destroyed agricultural yields and water infrastructure, and agricultural inputs disrupted in foreign markets. The conflict is also affecting the ability of international agencies to provide food aid to countries that are suffering from famine or other armed conflicts [70]. An energy price increase will also influence energy-related inputs (fertilizer, pesticides), which is likely to have worldwide impacts on agricultural production, and as a result, will affect agricultural product prices that are also being influenced by prewar factors (e.g., COVID-19) [71]. There is a strong connection to cluster C4 (economics and public policies) that illustrates the impact of war on various aspects of the economy and vice versa.

Despite the assumption that there is enough food on the global level, higher food prices could become a problem and put food availability under some pressure for the part of the population that has a low income, spends a large part of their food on cereals, and is highly dependent on imports of Ukrainian and Russian cereals (e.g., Egypt, Turkey, Lebanon, Tunisia, Yemen, Syria, Afghanistan, Nigeria) [69,70,74,77]. This war is also affecting the ability of international agencies to provide food aid to countries that are suffering from famine or other armed conflicts [70].

Based on the research, solutions to the listed problems can be suggested. One such solution is an improvement in decision making by policymakers and private parties to minimize war-induced losses by forecasting food production using satellite imagery [72]. Carefully selected structural components of the national program of balanced production of food (e.g., increasing the level of income, providing cheap food, increasing the supply of food, the use of special tools) can allow for solutions for the tasks at the local level and impact sustainable agricultural development, which can influence global food security [73]. On the global level, we can provide a reduction in food waste and accelerate the shift toward healthier diets with fewer animal products (including increased production of legumes) [74,76]. At the same time, a change in the existing model of global trade is suggested by [74] in order to increase food stocks in countries that have disinvested in domestic production in favor of cheaper imports. This is also in line with the strengthened Farm2Fork conception by [76]. On the other hand, the authors of [75] have different opinions on this question. They outline that those transitions to (more) closed food economies are not advisable, as this would remove players from international markets, could lead to food shortages in many countries, and might take focus away from environmental and health-related issues. The authors suggest a reduction in bureaucratic and tariff barriers to trade by global supply chain structures. In order to protect food systems-related infrastructures and activities, the authors of [70] propose that the international community establish a strategic food reserve or include new rules in international humanitarian law. There is no doubt that systems of agricultural production need to be less energy-dense [74]. It relates to cluster C4 (economics and public policies) by showing how war shapes public policies in politics, health, food trade, and production.

The author of [44] argued that alongside classic responses such as trade controls, supply diversification, public support, and aid, the relatively comfortable position of Arab Gulf countries will play a crucial role in mitigating some of the impacts of food crises on the Middle East through regional cooperation and aid-related food security and fiscal stability. At the same time, local agriculture remains an important food security tool in some Middle Eastern countries, but it should be securitized using sustainability and efficiency criteria (particularly the issues of water availability and use efficiency). However, the aggravated food insecurity in some of the countries, particularly Yemen, as a result of the war in Ukraine is difficult to resolve in the short or medium terms. Despite having agricultural potential, decades of water over-abstraction, mismanagement, and cultivation of cash crops have left Yemen’s agricultural sector quite weak. The food crises in Middle Eastern countries will deteriorate the situation further if no lasting peace materializes and if donors as well as neighboring countries do not increase their aid [44].

The case studies’ results complement the findings from the bibliometric analysis, evidencing that those aspects such as food security and government, nutrition and war, food safety and health, and economics and public policies can also be found in the context of the war in Ukraine.

At the same time, the results provide a better understanding of the fact that the war in Ukraine also unleashed a new type of complex global food crisis with supply interruptions and rising costs of key agricultural inputs such as fuel, transport, and fertilizers. The results indicate that this war is not only generating food insecurity in Ukraine, but also globally, due to the shortage in grain production and supply and the higher prices.

## 4. Conclusions

This study aimed to explore the extent of the food insecurity problem due to the war in Ukraine by unveiling the research streams in which scholars have been conducting research related to the impact of war on food security aspects. Ten case studies that directly tackle the food security problem in the context of the war in Ukraine helped to reveal relevant secondary data that could support the bibliometric exploration, which could further contribute to policymaking.

The results from the bibliometric analysis indicate the existence of at least four main thematic clusters that scholars are covering on the food security aspects in the general war context (not only in Ukraine): food security and government aspects (C1), human history, nutrition, and war (C2), agriculture, food safety, and health (C3), and economics and public policies (C4). The case studies complement the findings in these clusters, evidencing that these aspects can also be found in the context of the war in Ukraine, which is further evidenced by the secondary data gathered.

This study also has implications that the authors agree upon for their relevance for policy making. Firstly, the results indicate that the war is not solely generating food insecurity in Ukraine, but also globally, due to the shortage in grain production and the higher prices. Secondly, it has been evidenced that the war could compromise the capacity of food systems and supply chains. This could reduce the capacity of consumers to have access to sufficient amounts of food and important micronutrients, which could lead to malnutrition. Moreover, the shortage of food production can also be a consequence of the reduced import or production of fertilizers and agrichemicals. Finally, the situation unveiled the necessity of consistent food policies to tackle the situation on a local and global scale, especially related to the necessity of sustainable food systems in order to guarantee the health of people and the planet.

This study has limitations which could be related to the methods deployed. The bibliometric analysis based on the co-occurrence analysis operates under the assumption that the mere frequency of term co-occurrence signifies strong conceptual relationships, disregarding potential nuances in contextual meaning. Additionally, the quality and comprehensiveness of the data sources can impact the accuracy of results, potentially introducing biases or incomplete insights. The subjectivity involved in term selection and the dynamic nature of language further add complexities to the interpretation. Also, reliance on 10 expert-selected case studies may limit the generalizability of findings and introduce selection bias, potentially impacting the broader applicability and representativeness of conclusions. The authors believe, however, that these limitations could bring relevant ideas for future studies.

Firstly, the authors believe that gathering qualitative data through interviews with relevant stakeholders could contribute to elaborating the policymaking aspects previously suggested. In addition, a deep understanding of the reality of farmers in Ukraine and an exploration of their necessities could help the Ukrainian government to formulate specific policies to alleviate the problems related to food production and the food supply chain. Thirdly, an assessment conducted by the Ukrainian government to understand the health conditions of the Ukrainian citizens can help find those who need urgent support in terms of malnutrition, whether it involves the daily amount of food consumed or the important micronutrients. Fourthly, the authors believe that future research could seek to understand the current policies in place and find relevant gaps that have not been tracked by Ukrainian and European Union authorities in order to guarantee the proper functioning of food systems and the health of the planet and population at the same time.

## Figures and Tables

**Figure 1 foods-12-03996-f001:**
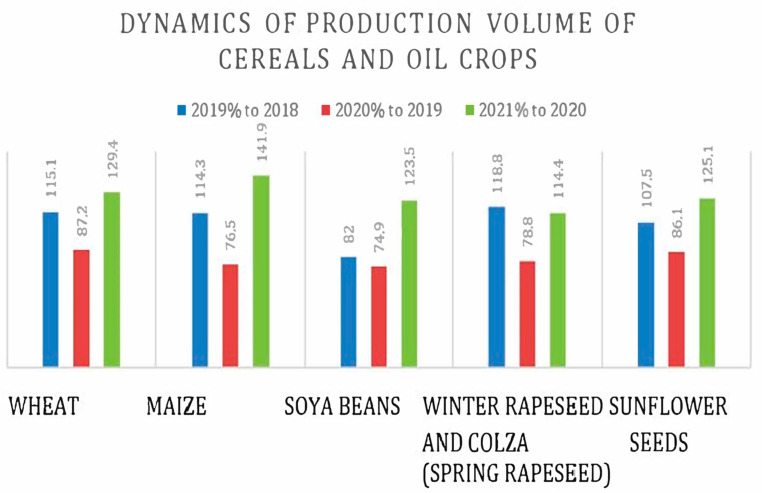
Dynamics of production volume of cereals and oil crops (the volume of production, yield, 2019, 2020, 2021).

**Figure 2 foods-12-03996-f002:**
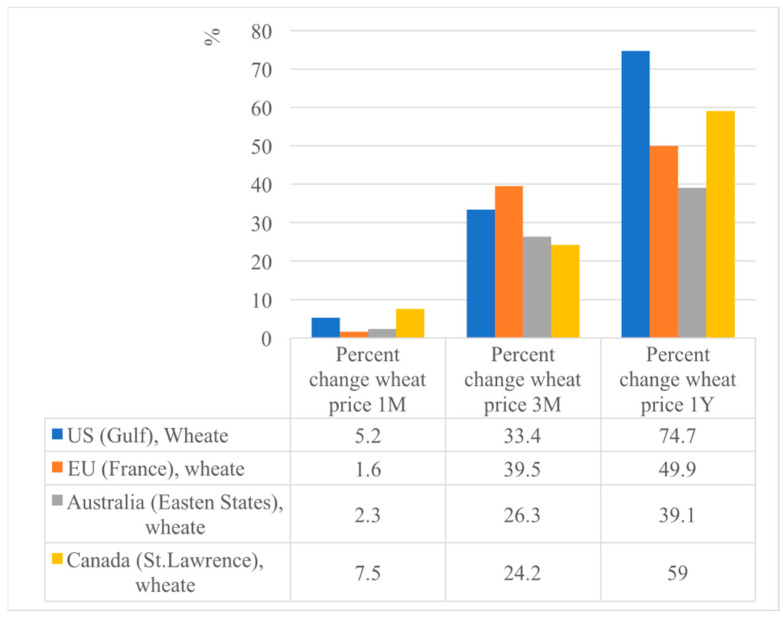
Percent change in wheat price from 1 month to 1 year (May 2021–May 2022) [2].

**Figure 3 foods-12-03996-f003:**
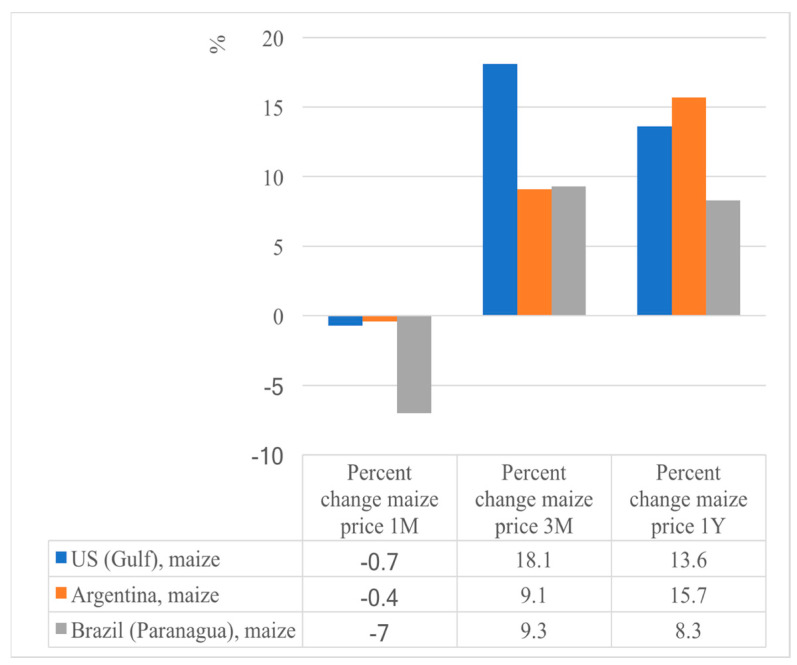
Percent change in maize price from 1 month to 1 year (May 2021–May 2022) (FAO) [2].

**Figure 4 foods-12-03996-f004:**
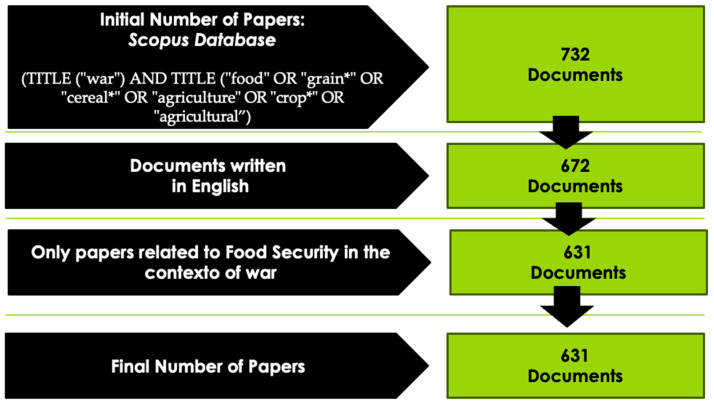
Stages of selecting articles for the bibliometric analysis.

**Figure 5 foods-12-03996-f005:**
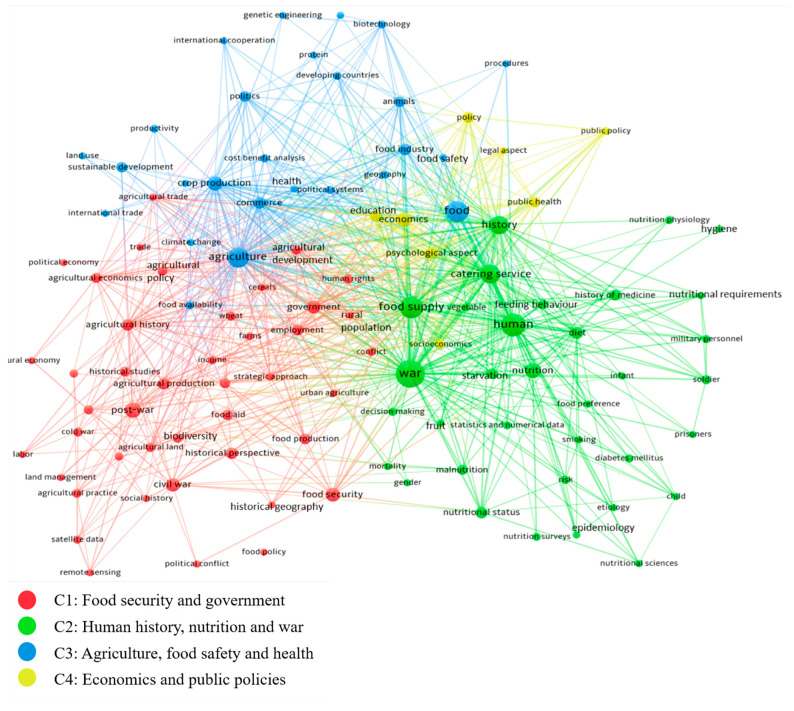
Results obtained by bibliometric analysis and division in clusters.

**Table 1 foods-12-03996-t001:** Harvesting of cereals and oil crops (the volume of production, yield, 2019, 2020, 2021). Data exclude the temporarily occupied territory of the Autonomous Republic of Crimea, the city of Sevastopol, and a part of temporarily occupied territories in the Donetsk and Luhansk regions [11].

All Agricultural Holdings Years	Harvesting of Volume of Production, Thousands of Tons (Yield, Tons per ha of the Harvested Area)				
	Wheat	Maize	Soybeans	Winter Rapeseed and Colza (Spring Rapeseed)	Sunflower Seeds
1.12.2021	32,719.67 0.464	39,819.37 0.801	3409.34 0.268	2960.08 0.295	16,439.84 0.252
1.12.2020	25,276.11 0.388	28,059.99 0.569	2770.93 0.209	2586.35 0.234	1313.8 0.206
1.12.2019	28,851.71 0.425	36,675.78 0.712	3698.71 0.232	3280.32 0.259	15,254.12 0.259
1.11.2018	25,070.92 0.382	25,516.52 0.743	4266.6 0.264	2780.67 0.270	13,882.71 0.234

**Table 2 foods-12-03996-t002:** Search criteria and the number of entries.

Database	Search String	Number of Documents
Scopus	(TITLE (“war”) AND TITLE (“food” OR “grain*” OR “cereal*” OR “agriculture” OR “crop*” OR “agricultural”)	Before screening process: 732 entries After screening process: 631 entries

Note: The asterisks correspond to the plural form of terms used.

**Table 3 foods-12-03996-t003:** Implications of the case studies on food security.

Case Study	Implications for Food Security
1. “Impacts of the conflict in Ukraine on global food security” [69].	Despite the fact that from a food security perspective, there is enough food on the global level, higher food prices put food availability under pressure for the low-income part of the population that is highly dependent on imports of Ukrainian and Russian cereals (for example, in Egypt, Turkey, and the Middle East).
2.”Implications of the Russia–Ukraine war for global food security” [70].	The solution to the rising global hunger fostered by the ongoing war in Ukraine could be establishing an international community strategic food reserve or including new rules in international humanitarian law that provide sufficient protection to food systems-related infrastructures and activities.
3. “Potential medium-term impacts of the Russia-Ukrainian war on the Dutch agriculture and food system: An assessment” [71].	An energy-related input (fertilizer, pesticides) price increase due to the war against Ukraine is likely to have worldwide impacts on agricultural production and consumption and will create a passing on effect on agricultural product prices.
4.”Quantifying War-Induced Crop Losses in Ukraine in Near Real Time to Strengthen Local and Global Food Security” [72].	Food production analyses by satellite imagery can help to improve decision making by policymakers and private parties to minimize war-induced agriculture losses.
5.”The conceptual principles of state policy of Ukraine in the field of food security in terms of European integration” [73].	The proposed strategic tools (structural, special, and local measures) enable solutions for nutrition improvement at the local level in Ukraine that at the same time also led to the development of sustainable agriculture, which can strengthen global food security.
6. “The reinvasion of Ukraine threatens global food supplies” [74].	In order to combat food insecurity, especially in vulnerable countries (Lebanon, Yemen, Syria, Afghanistan), it is suggested to replace all animal-based foods with plant-based ones, to increase food stocks in countries that have disinvested in domestic production in favor of cheaper imports, and to make agricultural production less energy-dense.
7. “The War in Ukraine, Agricultural Trade and Risks to Global Food Security” [75].	The key to dealing with mitigating the risks of food shortages increased by the war against Ukraine for import-dependent countries like the MENA region and sub-Saharan Africa could be a reduction in bureaucratic and tariff barriers to trade by global supply chain structures. And transitions to (more) closed food economies could lead to food shortages in many countries.
8. “We need a food system transformation—In the face of the Russia-Ukraine war, now more than ever” [76].	Three levers for solving short-term problems of food insecurity and long-term sustainable development: accelerate the shift toward healthier diets with fewer animal products in Europe, increase the production of legumes, and strengthen Farm2Fork and reduce the amount of food waste.
9. “What the war in Ukraine means for energy, climate and food” [77].	It outlines the possibility of rising energy prices and the potential loss of grain supplies from Ukraine and Russia to reinforce inflationary effects and drive up prices for food and other commodities for people who are already severely struggling (Yemenis, Syrians, and Nigerians).
10. “Caught off guard and beaten: The Ukraine war and food security in the Middle East” [44].	The war in Ukraine has had varying impacts in the Middle East countries (especially Yemen, Libya, Lebanon, and Sudan), including a deepened food sector crisis, worsened by political–economic instabilities, limited domestic agriculture, and the lack of reliable grain reserves. At the same time, there are country-level response strategies like food subsidy systems and regional aid and cooperation that have emerged in the Gulf countries to mitigate impacts.

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
