# Peer review of "How the War in Ukraine Affects Food Security"

_foods, 2023, doi:10.3390/foods12213996_

Round 1

Reviewer 1 Report

Comments and Suggestions for Authors

Honestly, your study seems a bit confusing. Starting with the introduction, which seems very long to me (by the way, references are not cited that way).
In the methodology section, it would be interesting to put a flow chat of how you worked and how you discriminated papers (I recommend you see the flow chart in this paper: https://doi.org/10.3390/agriculture13020340). It is also not clear how many final papers you selected ... it doesn't seem like much of a literature review to me. But what puzzles me most is the selection of the 10 case studies. You have basically summarized what has been said and written by others and commented on it arbitrarily...

Author Response

Thank you very much for taking the time to review this manuscript. Please find the detailed responses below and the corresponding revisions highlighted in the re-submitted files.

Comment

Response

Honestly, your study seems a bit confusing. Starting with the introduction, which seems very long to me (by the way, references are not cited that way).

Thank you for the comment. The authors have made their best efforts to address all comments. Here, the size of the introduction seems appropriate since it provides a review of the literature. The references have been adjusted to fit with the style of the journal.

In the methodology section, it would be interesting to put a flow chart of how you worked and how you discriminated papers (I recommend you see the flow chart in this paper: https://doi.org/10.3390/agriculture13020340). It is also not clear how many final papers you selected ... it doesn't seem like much of a literature review to me. 

Thank you for the comment. We have added a flow chart to show how we have selected the papers. We also state more clearly how many papers were selected.

But what puzzles me most is the selection of the 10 case studies. You have basically summarized what has been said and written by others and commented on it arbitrarily...

Thank you for the comment. The case studies approach is a technique used to exemplify how some trends are being approached. The interpretation is not arbitrary. Rather, it draws lessons from the case studies. We have adjusted the text to make it clearer the criteria for the selection of the case studies: 

thematic relevance, variety of scopes, and international interest.

Reviewer 2 Report

Comments and Suggestions for Authors

Dear Authors,

the conditions under which humanity operates in the Anthropocene are radically different from before. We are running a predatory economy destroying our planet and it was not until the outbreak of the COVID-19 pandemic that humans became convinced that the functioning of economies, globalised markets, complex supply chains, the pursuit of exorbitant GDP figures and the increase in transport and travel around the globe needed to be urgently reviewed. The conflict between Russia and Ukraine was another shock, as the problem of world food insecurity returned with redoubled force as a result of the war. And in this context, the reviewed manuscript should be considered important and necessary. However, I have some suggestions that might enhance its scientific quality.

The main suggestions concern: the introduction, the description of the clusters and the presentation of the case studies.

Introduction. I suggest that when addressing subsequent aspects, more details, facts and figures from recent reports should be added. For example, the current number of people who are hungry and do not have access to proper nutrition is given in the SOFI 2022 report [FAO, 2023]. Since in L. 35 and L. 41 are the first references to food security (FS), it would also be appropriate to define this concept at the beginning, and not only in L. 182. The four dimensions of FS are presented here, but there is no definition adopted by the World Food Summit 1996. Similarly, in L. 45 it is stated that food insecurity (FInS) affects 2.37 billion people, in L. 149 it is stated that 1/3 of the population of Ukraine is affected by FInS - how many million people is that? and in L. 163 mentions 600 million chronically malnourished. The authors should explain how the terms FInS, malnutrition, chronically undernourished are to be interpreted-do they use them as synonyms or does each mean something different?

In L. 59-70 it is more important to show the share of Ukraine alone in world exports, not together with Russia. In general, one should stick more to the issue of Ukraine. There is no need to include Figures 2 and 3, the description in L.120-124 is also superfluous, it should rather be indicated how the FAO food price index for cereals or the world prices of wheat, rapeseed etc. have increased.

L. 74 table 1 shows "volume of production" and does not show the volume of "yield". I propose to simplify this table - add a row above the row with column names, but without separating the columns and write there "Volume of production, thousands of tons"; leave as column names only: wheat, maize, soybean, rapessed, sunflower seeds; include reference 1 in the title of the table, since it applies to all columns; add the source of this data; in numerical values instead of a comma there should be a dot. In the commentary to this table, it is worth evaluating Ukrainian production relative to e.g. the EU, Europe, the world.

L. 82 the title of the chart needs improvement - it should be as in the chart window. And the bars for individual raw materials should have only the name of the raw material as the label name.

L. 87 I suggest listing the 10 largest importers here instead of referring readers to the source.

L. 92-93 superfluous sentence; a reference to the source is sufficient.

L. 156-157 the entry "the number of people suffering from malnutrition was expected to increase to 13.1 million globally due to this crisis" needs to be corrected as this is not a true figure.

Description of clusters. It should be written how many articles were assigned to each cluster; from the figure it can be seen that the least were related to cluster C4 'economics and public policies'. Perhaps the authors would be tempted to find the reasons, since both war and pandemic strongly determined the prices of food commodities on world and domestic markets. In my opinion, C2 also relates strongly to food supply. The source used in L. 298-301 refers to the nutritional aspects of FS, so it relates to C2 and should be credited here. On the other hand, the description of cluster C3, in addition to the relationships indicated, also relates strongly to environmental aspects, which should be reflected in its name. Still regarding cluster C4, it is worth noting the phenomenon of spoilage of the market for cereals and other agricultural raw materials in countries such as Poland, Hungary or Romania due to the import of raw materials from Ukraine (they have lower prices than in these countries because labour is cheaper) and the EU trade policy measures to protect the EU market.

Case studies (CS). State what criteria the experts used to select these 10 articles as CSs. I propose to make a table in the summary of the presentation of these CSs, which collects the implications for the CSs because of the ongoing war in Ukraine. In L. 574 it should be explained what Farm to Fork is. Among these CSs, CS8 is the most interesting, because the authors presented the only possible positive of this war (if that can be said at all). They indicated the need to accelerate the transformation of food systems, pointing to 3 specific actions. If there is less supply of soya, maize, rapeseed meal in the world, the world will be forced to reduce the production of raw materials of animal origin, they will become more expensive and then people in rich countries will be quicker to change their diets to one based on plants, including pulses, and their diets will become more sustainable and healthy. Of course, this conflicts with the economic goals of these countries and their policies to increase each agricultural production. I suggest that the Authors of the manuscript address this issue in the summary of the CSs. Finally, a minor editorial suggestion - the titles of the CSs are article titles and should therefore be written in inverted commas.

Regarding the manuscript as a whole - it is necessary to correct the records of reference to sources according to the requirements of the journal editors (numbering according to the order of appearance).

Other minor suggestions:

L. 240 under the table should be written what the asterisks in column 2 mean.

L. 244 the correct chart number needs to be written.

L. 269 mistake - not 'Stations' but 'Nations'.

L. 615 sentence "According to the..." should start with a new paragraph.

Kind regards

Comments on the Quality of English Language

The quality of English language is good, I have pointed out a few corrections.

Author Response

Thank you very much for taking the time to review this manuscript. Please find the detailed responses below and the corresponding revisions highlighted in the re-submitted files.

INTRODUCTION. I suggest that when addressing subsequent aspects, more details, facts and figures from recent reports should be added. For example, the current number of people who are hungry and do not have access to proper nutrition is given in the SOFI 2022 report [FAO, 2023]. Since in L. 35 and L. 41 are the first references to food security (FS), it would also be appropriate to define this concept at the beginning, and not only in L. 182. The four dimensions of FS are presented here, but there is no definition adopted by the World Food Summit 1996. Similarly, in L. 45 it is stated that food insecurity (FInS) affects 2.37 billion people, in L. 149 it is stated that 1/3 of the population of Ukraine is affected by FInS - how many million people is that? and in L. 163 mentions 600 million chronically malnourished. The authors should explain how the terms FInS, malnutrition, chronically undernourished are to be interpreted-do they use them as synonyms or does each mean something different?

Thank you for the comment. We have made the following changes:

  1. Added a definition of food security based on the FAO and World Food Summit
  2. Added more 3 references and details on works on poor nutrition
  3. We provide details on how many people in Ukraine are exposed to food insecurity 

We use the words  “chronically malnourished” for those semi-permanently suffering from malnutrition, so the meanings are different. 

In L. 59-70 it is more important to show the share of Ukraine alone in world exports, not together with Russia. In general, one should stick more to the issue of Ukraine. There is no need to include Figures 2 and 3, the description in L.120-124 is also superfluous, it should rather be indicated how the FAO food price index for cereals or the world prices of wheat, rapeseed etc. have increased.

Thank you for the comment. We have changed the text to show the share of Ukraine exports without Russia.  We prefer to keep Figures 2 and 3 since these are useful to the international audience. We removed the description in lines 120-124 and now included a text on how food prices have increased.

L. 74 table 1 shows "volume of production" and does not show the volume of "yield". I propose to simplify this table - add a row above the row with column names, but without separating the columns and write there "Volume of production, thousands of tons"; leave as column names only: wheat, maize, soybean, rapessed, sunflower seeds; include reference 1 in the title of the table, since it applies to all columns; add the source of this data; in numerical values instead of a comma there should be a dot. In the commentary to this table, it is worth evaluating Ukrainian production relative to e.g. the EU, Europe, the world.

Thank you for the comment. We have made the changes to Table 1 suggested but were unable to find reliable literature on Ukraine production relative to the EU or Europe since there is no official benchmarking. 

L. 82 the title of the chart needs improvement - it should be as in the chart window. And the bars for individual raw materials should have only the name of the raw material as the label name.

L. 87 I suggest listing the 10 largest importers here instead of referring readers to the source.

L. 92-93 superfluous sentence; a reference to the source is sufficient.

L. 156-157 the entry "the number of people suffering from malnutrition was expected to increase to 13.1 million globally due to this crisis" needs to be corrected as this is not a true figure.

Thank you for the comment. We have adjusted the title of the chart to fit with the chart window. We also adjusted the bar for raw materials, as suggested.

Thank you for the comment. We have now listed the 10 large importers.

Thank you for the comment. We have removed the sentence.

Thank you for the comment. We have now added a more accurate figure and the reference to support it. 

DESCRIPTION OF CLUSTERS. It should be written how many articles were assigned to each cluster; from the figure it can be seen that the least were related to cluster C4 'economics and public policies'. Perhaps the authors would be tempted to find the reasons, since both war and pandemic strongly determined the prices of food commodities on world and domestic markets.

Thank you for the comment. We understand your concern in identifying how many articles were assigned to each one of the clusters. However, providing a precise number of papers for each cluster obtained in VOSviewer software based on the co-occurrence of terms is challenging since this technique is valuable for uncovering thematic patterns in scientific literature, but it falls short of yielding precise counts for several reasons. Firstly, the approach relies on the occurrence of terms within documents, and the same term can be used in different contexts or may have varying degrees of relevance to a paper's main theme. This ambiguity in term usage makes it difficult to definitively assign papers to clusters. Secondly, co-occurrence analysis often necessitates setting arbitrary thresholds for term co-occurrence, impacting which papers are included in clusters and introducing subjectivity into the process. Finally, scientific papers frequently span multiple topics, leading to papers being allocated to multiple clusters, thus complicating precise counting.

The case studies’ results complement the findings from the bibliometric analysis, particularly by finding the reasons related to cluster C4 'economics and public policies' in the context of the war in Ukraine. We have added an additional sentence to reflect the connections between cluster 4 in the case study's summary.

In my opinion, C2 also relates strongly to food supply. The source used in L. 298-301 refers to the nutritional aspects of FS, so it relates to C2 and should be credited here. On the other hand, the description of cluster C3, in addition to the relationships indicated, also relates strongly to environmental aspects, which should be reflected in its name. Still regarding cluster C4, it is worth noting the phenomenon of spoilage of the market for cereals and other agricultural raw materials in countries such as Poland, Hungary or Romania due to the import of raw materials from Ukraine (they have lower prices than in these countries because labour is cheaper) and the EU trade policy measures to protect the EU market.

Thank you for the comment. We have added an additional text to reflect the connections between the clusters.  We also added a sentence on the market spoilage when describing C4.

CASE STUDIES (CS). State what criteria the experts used to select these 10 articles as CSs.

Thank you for the comment. We have adjusted the text to make it clearer the criteria for the selection of the case studies:  thematic relevance, variety of scopes, and international interest.

I propose to make a table in the summary of the presentation of these CSs, which collects the implications for the CSs because of the ongoing war in Ukraine. In L. 574 it should be explained what Farm to Fork is. Among these CSs, CS8 is the most interesting, because the authors presented the only possible positive of this war (if that can be said at all). They indicated the need to accelerate the transformation of food systems, pointing to 3 specific actions. If there is less supply of soya, maize, rapeseed meal in the world, the world will be forced to reduce the production of raw materials of animal origin, they will become more expensive, and then people in rich countries will be quicker to change their diets to one based on plants, including pulses, and their diets will become more sustainable and healthy. Of course, this conflicts with the economic goals of these countries and their policies to increase each agricultural production. I suggest that the Authors of the manuscript address this issue in the summary of the CSs. Finally, a minor editorial suggestion - the titles of the CSs are article titles and should therefore be written in inverted commas.

Thank you for the comment. We have added a table to summarise the CCs. We also explain that the “Farm to Fork” is, namely a scheme led by the European Union, to promote the sustainability of farming processes, i.e. from production (the farm) until it reaches the plate of the end-consumer (the fork).

Further to the suggestion, we added a sentence to the summary of the case studies:

“It should be noted that the case studies illustrate a variety of situations and contexts where food security is endangered or is avoided, and the various means used to achieve it”.

Finally, we now use inverted commas in the titles of the case studies.

Regarding the manuscript as a whole - it is necessary to correct the records of reference to sources according to the requirements of the journal editors (numbering according to the order of appearance).

Thank you for the comment. We have revised all references, made sure they are accurate, and ensured their numbering follows the journal´s style.

L. 240 under the table should be written what the asterisks in column 2 mean.

Thank you for the comment.  We added an explanation for the asterisks under the table.

L. 244 the correct chart number needs to be written.

Thank you for the comment. We have added the charter number.

L. 269 mistake - not 'Stations' but 'Nations'.

L. 615 sentence "According to the..." should start with a new paragraph.

Thank you for the comment. We have corrected it.

Thank you for the comment. We have corrected it.

Kind regards,

Authors team

Reviewer 3 Report

Comments and Suggestions for Authors

This is an especially relevant issue since at this time there are a multitude of armed conflicts on the planet with the consequent suffering in general and lack of access to food for the population.

The conflict in Ukraine is of special relevance since it not only affects the availability of food for its population but also has an impact on the food availability of basic products in other countries, especially impoverished countries. The methodology used to achieve the proposed objectives is innovative. The results obtained are interesting.

Methodology:

The search strategy, the search string and the terms used are very well explained. However, it is striking that the 631 articles used in the review are not JCR. Is there a reason why the authors have chosen Scopus?

On the other hand, beyond the search terms they do not explain what criteria are used when selecting the papers, language, data collection methodology, type of data, etc. Furthermore, the authors explain that the VOSviewer software takes into account only the search terms (line 247, page 7). I don't know the software, but in my opinion, the authors should do a prior screening, selecting articles that meet sufficient quality criteria in their results.

In my case, I am not familiar with VOSviewer software, I would have appreciated a more detailed explanation of how clusters are generated.

In relation to the 10 case studies, are they papers other than those included in the VOSviewer? The authors comment that the case study selection was based on the expert-driven literature review to find relevant well-cited papers that are published in top journals (lines 258-260, page 7). Could you explain better how this expert-driven literature review is carried out?

Results

Lines 84 and 85 of page 3. Place figure 2 below that sentence for better understanding of the text.

Line 126. Figure 2. The text on the X axis is not visible. Perhaps there would be a table underneath, as in figure 3.

Perhaps the 4 clusters obtained with the software are not so clear, I clearly see 3, however C1 and C4 seem the same. In fact, explanations of the results of the clusters are repeated, for example part of the explanation from C3 has also been used in C2.

In section 3.2. Food security in conflict context in Ukraine, I think that it would be better to put the  citation of the article next to the title of each case, instead of putting it at the end of the explanation of the case.

Author Response

Thank you very much for taking the time to review this manuscript. Please find the detailed responses below and the corresponding revisions highlighted in the re-submitted files.

The search strategy, the search string and the terms used are very well explained. However, it is striking that the 631 articles used in the review are not JCR. Is there a reason why the authors have chosen Scopus?

Thank you for your comment. The authors chose to use Scopus over Web of Science due to its broader coverage of international and multidisciplinary sources, advanced citation tracking capabilities and specific database features that align better with their research objectives. In this sense, Scopus's diverse and extensive content and user-friendly features made it a more suitable choice for their study.

On the other hand, beyond the search terms they do not explain what criteria are used when selecting the papers, language, data collection methodology, type of data, etc. Furthermore, the authors explain that the VOSviewer software takes into account only the search terms (line 247, page 7). I don't know the software, but in my opinion, the authors should do a prior screening, selecting articles that meet sufficient quality criteria in their results.In my case, I am not familiar with VOSviewer software, I would have appreciated a more detailed explanation of how clusters are generated.

Thank you for the comment. We have explained more clearly the criteria used, also by adding a new figure (Figure 4) and added some information on how the clusters are generated below Table 2.

In relation to the 10 case studies, are they papers other than those included in the VOSviewer? The authors comment that the case study selection was based on the expert-driven literature review to find relevant well-cited papers that are published in top journals (lines 258-260, page 7). Could you explain better how this expert-driven literature review is carried out?

Thank you for the comment. We have adjusted the text to make it clearer the criteria for the selection of the case studies:  thematic relevance, variety of scopes, and international interest. These are decoupled from the bibliometric analysis, which is a second methodology used in the study.

Lines 84 and 85 of page 3. Place figure 2 below that sentence for better understanding of the text.

Thank you for the comment. We have adjusted the text as suggested.

Line 126. Figure 2. The text on the X axis is not visible. Perhaps there would be a table underneath, as in figure 3.

Thank you for the comment. We have adjusted the text as suggested.

Perhaps the 4 clusters obtained with the software are not so clear, I clearly see 3, however C1 and C4 seem the same. 

Thank you for the comment. We have adjusted the text as suggested to make it clearer. Cluster C1 (red cluster) primarily delves into the social dimensions of war's impact on food security, emphasizing displacement, veterans' experiences, and the consequences of trade wars, with a focus on nutritional aspects. In contrast, Cluster C4 (yellow cluster) centers around the economic consequences of war and political conflict, discussing how economic struggles, market failures, and armed conflicts are interconnected and impact not only food security but also public policies across various domains. While both clusters touch on aspects of war's effects, they differ in their primary focuses and depth of coverage, with Cluster C1 emphasizing social and nutritional aspects and Cluster C4 emphasizing economic and policy dimensions.

In fact, explanations of the results of the clusters are repeated, for example part of the explanation from C3 has also been used in C2.

Thank you for the comment. We have adjusted the text so that duplications are now avoided.

In section 3.2. Food security in conflict context in Ukraine, I think that it would be better to put the  citation of the article next to the title of each case, instead of putting it at the end of the explanation of the case.

Thank you for the comment. We have adjusted the text as suggested and put the citation now in the beginning.

Kind regards,

Authors team

Round 2

Reviewer 1 Report

Comments and Suggestions for Authors

Now the paper is publishable

Reviewer 2 Report

Comments and Suggestions for Authors

Dear authors,
I am satisfied with both the corrections in the text and your responses to my suggestions. By taking into account the reviewers' suggestions, the scientific value of the manuscript has been increased.
I have no further substantive comments, but I do have two linguistic comments:
1 - in Figure 2, 'wheat' is misspelled four times in the chart window,
2 - in the sentence in L. 83 the verb 'are' is missing.

Kind regards

Comments on the Quality of English Language

The linguistic quality of the manuscript is correct. I have only pointed out two linguistic errors above.